# *Aspects are Anchors*: Towards Multimodal Aspect-based Sentiment Analysis via Aspect-driven Alignment and Refinement

Zhanpeng Chen*
Peking University
Shenzhen, China
troychen927@stu.pku.edu.cn

Zhihong Zhu*
Peking University
Shenzhen, China
zhihongzhu@stu.pku.edu.cn

Wanshi Xu
Peking University
Shenzhen, China
xwanshi@stu.pku.edu.cn

Yunyan Zhang
Jarvis Research Center, Tencent
YouTu Lab
Shenzhen, China
yunyanzhang@tencent.com

Xian Wu[†]
Jarvis Research Center, Tencent
YouTu Lab
Shenzhen, China
kevinxwu@tencent.com

Yefeng Zheng
Medical Artificial Intelligence Lab,
Westlake University &
Jarvis Research Center, Tencent
YouTu Lab
China
zhengyefeng@westlake.edu.cn&
yefengzheng@tencent.com

## Abstract

Given coupled sentence image pairs, Multimodal Aspect-based Sentiment Analysis (MABSA) aims to detect aspect terms and predict their sentiment polarity. While existing methods have made great efforts in aligning images and text for improved MABSA performance, they still struggle to effectively mitigate the challenge of the noisy correspondence problem (NCP): the text description is often not well-aligned with the visual content. To alleviate NCP, in this paper, we introduce Aspect-driven Alignment and Refinement (ADAR), which is a two-stage coarse-to-fine alignment framework. In the first stage, ADAR devises a novel *Coarse-to-fine Aspect-driven Alignment Module*, which introduces Optimal Transport (OT) to learn the coarse-grained alignment between visual and textual features. Then the adaptive filter bin is applied to remove the irrelevant image regions at a fine-grained level; In the second stage, ADAR introduces an *Aspect-driven Refinement Module* to further refine the cross-modality feature representation. Extensive experiments on two benchmark datasets demonstrate the superiority of our model over state-of-the-art performance in the MABSA task.

## CCS Concepts

• **Information systems** → **Sentiment analysis**; **Multimedia and multimodal retrieval**.

## Keywords

Multimodal Aspect-based Sentiment Analysis, Optimal Transport

---

*Equal contribution.
[†]Corresponding author.

**ACM Reference Format:**
Zhanpeng Chen, Zhihong Zhu, Wanshi Xu, Yunyan Zhang, Xian Wu, and Yefeng Zheng. 2024. *Aspects are Anchors*: Towards Multimodal Aspect-based Sentiment Analysis via Aspect-driven Alignment and Refinement. In *Proceedings of the 32nd ACM International Conference on Multimedia (MM '24), October 28–November 1, 2024, Melbourne, VIC, Australia.* ACM, New York, NY, USA, 9 pages. https://doi.org/10.1145/3664647.3681189

## 1 Introduction

Multimodal aspect-based sentiment analysis (MABSA) has attracted increasing attention in recent years [9, 16, 29]. Typically, MABSA includes three downstream tasks: Multimodal Aspect Term Extraction (MATE), Multimodal Aspect-oriented Sentiment Classification (MASC), and Joint Multimodal Aspect-Sentiment analysis (JMASA) [12, 20, 27]. Specifically, MATE aims to identify aspect terms from a text-image pair [39]; MASC aims to classify the sentiment of each aspect term [34]; while JMASA aims to jointly extract the aspect terms and predict their sentiments. Therefore, JMASA refers to the end-to-end solution combining MATE and MASC. For example, as depicted in Figure 1, the target of JMASA is to analyze the multimodal information and output the aspect-polarity pairs *(Klay Thompson, POS), (Warriors, POS)* and *(TrailBlazers, NEG)*, while MATE and MASC only focus on identifying the aspect terms and classifying the sentiment, respectively.

Existing works [17, 38] propose to align the global image feature with the text feature to derive cross-modality feature representation. Despite the performance improvement, they ignore the fact that only some specific visual regions are relevant to the aspect terms, and vice versa. Therefore, the information simply extracted from the global features could introduce the noise correspondence problem (NCP) [11], thereby limiting further performance improvement. Motivated by this, the recent work AoM [41] firstly extracted the aspects from the sentence, and achieved aspect-guided cross-modal interaction to alleviate the NCP between image and text. Although AoM has achieved improved performance, it still suffers from two limitations: (i) Dependency on pre-trained tools: AoM

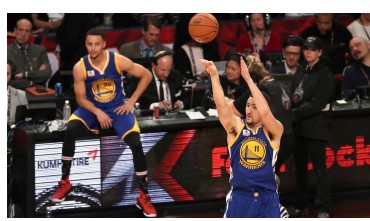

**Text**
Klay Thompson Warriors
overwhelm TrailBlazers
110-99 go up 2-0 in series
# NBAPlayoffs.

**Aspects & Sentiment**
(Klay Thompson , POS)
(Warriors, POS)
(TrailBlazers, NEG)

**Figure 1: An example of the MABSA task from Twitter2017 [34], including the text, aspects and sentiments.**

highly depends on the pre-trained phrase parsing tool Spacy[1] for noun extraction. However, the model fails when it comes to extracting nouns that are not well pre-trained. (ii) Noisy correspondence misalignment: AoM may mistakenly align aspect terms with irrelevant visual elements because the relevant ones are missing in the visual information, resulting in noisy correspondence misalignment. Take the case in Figure 1 for example, the player in the middle contributes to the correct detection of positive sentiment towards 'Klay Thompson' and 'Warriors'. However, the negative aspect term 'TrailBlazers' is not shown in the figure, which leads to the failure of detection.

To effectively handle the challenge of NCP while avoiding the aforementioned drawbacks, we propose Aspect-driven Alignment and Refinement (ADAR). As shown in Figure 3, ADAR includes two core modules: *Coarse-to-fine Aspect-driven Alignment Module* (ADAM) and *Aspect-driven Refinement Module* (ADRM). **We opt for Optimal Transport (OT) at the coarse-grained level instead of pre-trained tools to address the first limitation.** As a critical component in many downstream applications [1, 3, 38], OT has been employed to facilitate cross-modal retrieval tasks involving matching images with text descriptions and vice versa. As depicted in Figure 2, by computing pairwise similarities and using transport strategies, OT provides a robust tools-free framework to align between visual and textual representations considering the potential aspects as anchors. In ADAM, we first establish a flexible coarse-grained alignment between image and text by optimizing the goal of OT. To mitigate NCP, we directly project the visual representations towards textual feature space by employing the barycenter-based strategies [7], while previous work [22, 38] solve the problem by optimizing different loss functions.

**To address the second limitation, based on the output of OT, we further introduce an adaptive filter bin that deals with the case of noisy correspondence misalignment at the fine-grained level.** With the adaptive filter bin activated, each visual aspect can be aligned with relevant textual aspects or discarded based on a pre-selected threshold, leading to a precise alignment of existing visual elements instead of noisy correspondence misalignment. Subsequently, in the ADRM module, textual representations that already emphasize aspects can implicitly reason the complementary relations between images through meticulously crafted cross-modal attention, leading to accurate predictions.

[1]https://spacy.io/

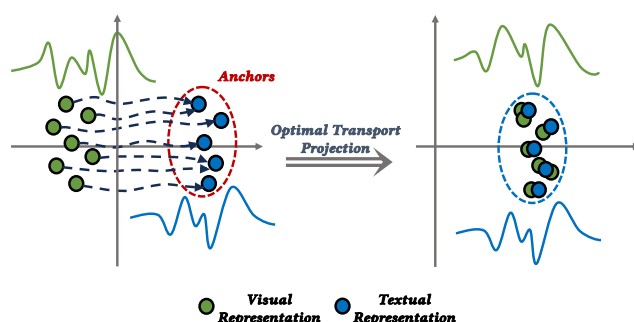

**Figure 2: An illustration of the optimal transport for multimodal representations. Green and blue curves represent distributions of different modal entities while corresponding dots represent the embeddings.**

Extensive experiments conducted on two benchmarks across three tasks demonstrate the effectiveness and robustness of ADAR. The comprehensive ablation analysis reveals how the proposed modules and methods complement each other. For instance, ADAM achieves a coarse-to-fine alignment by coupling the OT method and the adaptive filter bin. Furthermore, our method demonstrates superior performance compared to LLMs on MASC. Overall, our main contributions in this work are three-fold:

- We present a novel two-stage framework to address NCP under the guidance of aspects. To the best of our knowledge, this is the first work that introduces Optimal Transport to address NCP in MABSA.
- We design a coarse-grained aspect-driven alignment module with a fine-grained adaptive filter bin mechanism and an aspect-driven refinement module to effectively align and refine features between images and texts.
- Extensive experiments on two benchmark datasets show that ADAR significantly outperforms previous models. Further analysis verifies the advantages of our model.

## 2 Related Work

**Multimodal Aspect-based Sentiment Analysis**  As social media becomes increasingly rich with multimodal user posts, researchers have discovered that images provide valuable supplementary information for aspect term extraction [2] and sentiment analysis [29, 40]. Consequently, Multimodal Aspect-based Sentiment Analysis (MABSA) has garnered widespread attention and study. Within MABSA, two distinct sub-tasks have emerged: Multimodal Aspect Term Extraction (MATE) and Multimodal Aspect-oriented Sentiment Classification (MASC). MATE focuses on extracting all relevant aspect terms from a sentence with cues from the accompanying image, while MASC is concerned with predicting the sentiment polarities associated with these aspects.

In MABSA, which targets the nuanced analysis of fine-grained entities and their sentiments within a post, models are challenged to accurately apply pertinent visual information to the targeted aspect of each pair. To address cross-modal alignment, Ling et al. [17] developed a generative multimodal architecture grounded in BART. This architecture is designed for both vision-language pre-training and

downstream MABSA tasks. Meanwhile, Yang et al. [33] introduced a dynamic approach for controlling the influence of visual information on different aspects. This approach is based on the principle that the lower the confidence in text-only predictions, the greater the reliance on visual inputs. Furthermore, Zhou et al. [41] crafted an aspect-aware attention module coupled with a Graph Neural Network, which is aimed at discerning aspect-relevant multimodal content, considering both semantic and sentiment perspectives.

While existing methods make significant strides, they tend to overlook the misalignment between the visual and textual features. This oversight can lead to the introduction of irrelevant noise during the modeling process, as these methods lack fine-grained aspect-driven alignment.

**Optimal Transport**    Originally, Optimal Transport was introduced to quantify the distance between two probability distributions. Its application has recently garnered considerable attention across various domains, such as sequence alignment [19], domain adaptation [31] and document matching [37]. These applications highlight OT's capability in handling structured data and incorporating additional structural information beyond classic formulations. This has proven particularly useful in NLP tasks [1, 3, 25, 38] like unsupervised word translation, sentence similarity, domain adaptation, ontology alignment, etc. However, it's noteworthy that while the potential of OT is significant, its application in aligning images and text specifically for MABSA presents a novel challenge. This is because MABSA requires not just traditional modalities alignment but also a focus on aspect-driven alignment.

In this work, we propose a novel aspect-driven framework for MABSA. This framework comprises two key components: the Coarse-to-fine Aspect-driven Alignment Module and the Aspect-driven Refinement Module. The former is designed to align modalities through optimal transport, ensuring that the features align closely with the relevant aspects. The latter is aimed at effectively enhancing the interaction between the different modalities, ensuring a more coherent and integrated multimodal modeling.

## 3 Method

### 3.1 Preliminaries

**Task Definition**    Following previous work [17, 32, 41], we model the three downstream tasks on the same BART-based framework. Formally, when presented with a sample that encompasses an image denoted as $V$, and a sentence containing $n$ words, represented as $L = \{w_1, w_2, \ldots, w_n\}$, the outputs of JMASA, MATE, and MASC are formulated as:

- $Y_{JMASA} = \{a_1^s, a_1^e, s_1, \ldots, a_i^s, a_i^e, s_i, \ldots, a_k^s, a_k^e, s_k\}$,
- $Y_{MATE} = \{a_1^s, a_1^e, \ldots, a_i^s, a_i^e, \ldots, a_k^s, a_k^e\}$,
- $Y_{MASC} = \{a_1^s, a_1^e, s_1, \ldots, a_i^s, a_i^e, s_i, \ldots, a_k^s, a_k^e, s_k\}$,

where $a_i^s$, $a_i^e$, and $s_i$ denote the start index, the end index, and the sentiment polarity of the $i$-th aspect term in the tweet. The variable $k$ indicates the total number of aspects. Specifically when dealing with the MASC task, the start and end indexes are given during inference.

**Feature Extractor**    The initial word embeddings are derived from the pre-trained BART [14], selected for its exemplary capabilities

in textual representation. For the visual units, we employ preprocessing via ResNet [5] following Yu et al. [35].

## 3.2 Coarse-to-fine Aspect-driven Alignment Module

Despite the efficacy of the hidden states in capturing the characteristics of multimodal information, the correlations between different modalities remain relatively weak because of the noisy correspondence problem (NCP), thereby limiting performance. To address this, we design a coarse-to-fine module to align visual features to textual features by leveraging optimal transport with an adaptive filter bin mechanism.

**Coarse-grained Aspect-driven Transport**    To better utilize the contextual information, we first transfer the visual space to the textual space, driven by aspects from textual inputs. Let $H^V = \{h_1^V, \ldots, h_n^V\} \in \mathbb{R}^{n \times d}$ and $H^L = \{h_1^L, \ldots, h_m^L\} \in \mathbb{R}^{m \times d}$ denote the visual and textual hidden states extracted from the multimodal hidden states $H$, respectively. The distance between $h_i^V$ and $h_j^L$ is denoted as $S_{ij}$, which is an element in $S \in \mathbb{R}^{n \times m}$ and calculated with the Euclidean norm $S_{ij} = ||h_i^V, h_j^L||_2^2$. $T \in \mathbb{R}_+^{n \times m}$ denotes the transport assignment matrix where $T_{ij}$ represents the probability of aligning $h_i^V$ and $h_j^L$. Formally, the objective of OT is defined as follows,

$$OT(\mu, \nu, S) = \min_{T \in \Pi} \langle T, S \rangle, \tag{1}$$

where $\langle T, S \rangle = tr(T^\top S)$, $\Pi = \{T \in \mathbb{R}_+^{n \times m} \mid T1_m = \mu, T^\top 1_n = \nu\}$ and $1$ signifies an all-one vector. The terms $\mu$ and $\nu$ refer to the probabilistic simplex measures corresponding to $H^V$ and $H^L$, respectively. Considering that each image or text is independently sampled, we adopt equal weighting approach for each instance following Su and Hua [23], designated by $\mu = \frac{1}{n} 1_n$ and $\nu = \frac{1}{m} 1_m$. The optimal transport matrix, $T^*$, is then derived by minimizing the overall cost by Sinkhorn fixed point iterations [6],

$$T^* = \arg\min_{T \in \Pi} \langle T, S \rangle. \tag{2}$$

With the optimal transport matrix $T^*$ determined, we proceed to map $H^V$ to $\hat{H}^V$ for better utilizing the contextual information, which is facilitated by employing barycenter-based strategies,

$$\hat{H}^V = \text{diag}(1/\nu)((T^*)^\top + \Delta_T)H^V, \tag{3}$$

where $\Delta_T$ is an adjustable transport parameter [21]. To facilitate the projection of out-of-sample examples that were not encountered during the learning process of $T^*$, we initialize $\Delta_T$ using the Xavier Uniform distribution [8] with the same dimensions of $T^*$.

**Fine-grained Adaptive Filter Bin**    Although optimal transport is capable of aligning modalities effectively, texts and images in real-world scenarios often contain redundant or irrelevant information. Such noise can impede the efficacy of optimal transport in precisely aligning each visual unit with corresponding textual aspects. Motivated by Sarlin et al. [22], we introduce an innovative approach to noise reduction through the implementation of a fine-grained Adaptive Filter Bin (AFB), which is seamlessly integrated into the existing cost matrix $S_{ij}$ with simply one additional row and column,

$$\hat{S}_{i,m+1} = \hat{S}_{n+1,j} = \hat{S}_{n+1,m+1} = q, \tag{4}$$

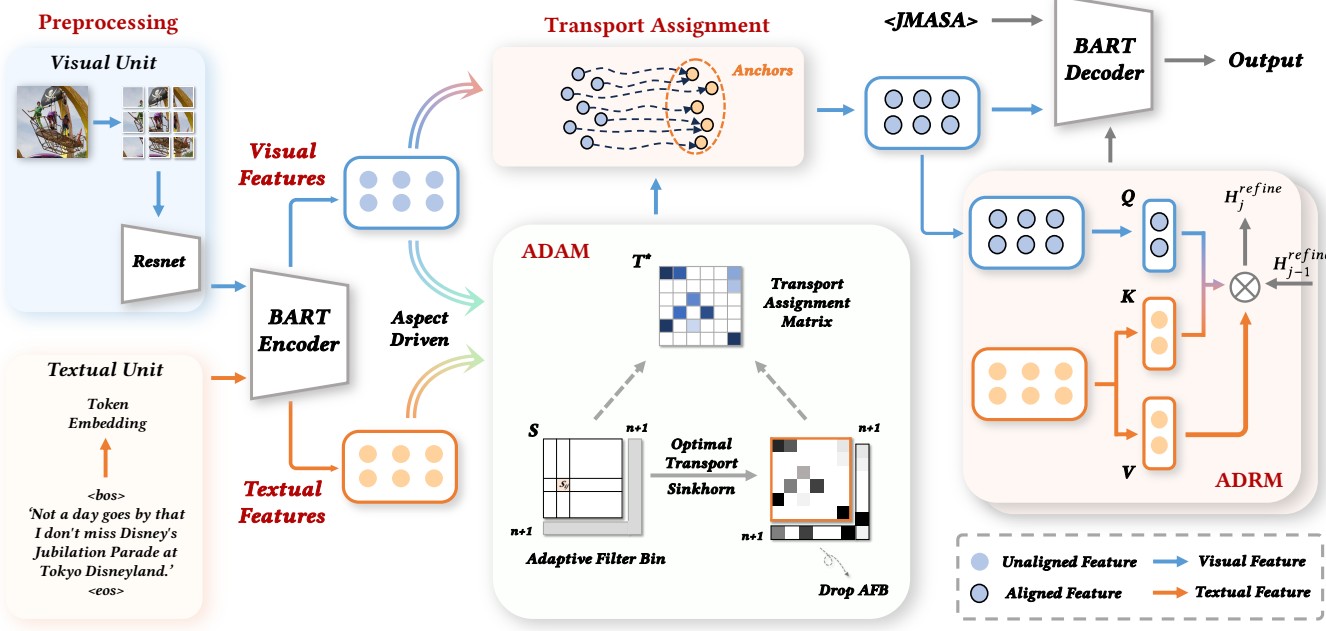

**Figure 3: The overview of our proposed ADAR framework. During generation, similar to Yan et al. [32], we insert a special token <bos> to indicate the beginning, and then insert a task-specific special token to indicate the task type. Specifically, <JMASA> informs the current task is JMASA, while <MATE> and <MASC> refer to MATE and MASC task, respectively.**

---

**Algorithm 1** Aspect-driven optimal transport

**INPUT:** Distribution $\mu$ and $\nu$ supported by visual representation $H^V$ and textual representation $H^L$, respectively.

**OUTPUT:** Aligned representation $\hat{H}^V$.

**INITIALIZE:** The size of visual representation $n$, the size of textual representation $m$ and the adjustable transport parameter $\Delta_T$.

1: $\mu = \frac{1}{n}\mathbf{1}_n$
2: $\nu = \frac{1}{m}\mathbf{1}_m$
3: $S_{ij} = \|\boldsymbol{h}_i^V - \boldsymbol{h}_j^L\|_2^2, \forall i \in [1, n], j \in [1, m]$
4: $T^* = \arg\min\langle T, S\rangle$ # Derived from the Sinkhorn Algorithm
5: $\hat{H}^V = \text{diag}(1/\nu)((T^*)^\top + \Delta_T)H^V$

---

where $\hat{S}_{i,j} = S_{i,j}$, $i \in [1, n]$ and $j \in [1, m]$. The hyper-parameter $q$ is strategically set to correspond to the top 20% cost associated with the original aligned image-text pairs. With the mechanism activated, each visual aspect can be either aligned with the relevant textual aspect or relegated to the bin. Consequently, $q$ effectively functions as a threshold, serving to exclude elements that are not alignable and thus maintaining a more noise-reduced environment for the execution of the transport assignment, i.e., $\hat{T}^* := \hat{T}_{1:n,1:m}$.

Now, we update the multimodal features as follows,

$$\hat{H}^{Vw} = H^V + \lambda_1 \hat{H}^V, \tag{5}$$

$$\hat{H} = [\hat{H}^{Vw}; H^L], \tag{6}$$

where $\lambda_1$ is the hyper-parameter to control the contribution from OT and $[;]$ is the concatenation operator for matrices.

## 3.3 Aspect-driven Refinement Module

Upon achieving effective alignment of feature distributions across various modalities through ADAM, the feature spaces still suffer from NCP and await a fine-grained interaction between visual and textual modality. To learn a refined cross-modality representation that contains relevance-suppressing information and complements language features, we introduce an Aspect-driven Refinement Module (ADRM) to enhance the learning of a refined multimodal representation.

Inspired by Liu et al. [18], we first utilize $\hat{H}_j^V$ as the query, and $H_j^L$ serves as both the key and value in this process. The interaction is conducted as follows,

$$
\begin{aligned}
\gamma_j &= softmax(\frac{Q^V(K^L)^\top}{\sqrt{d_k}}) \\
&= softmax(\frac{\hat{H}_{j-1}^V W_{Q^V}(H_{j-1}^L W_{K^L})^\top}{\sqrt{d_k}}),
\end{aligned} \tag{7}
$$

where $j \in \{1, 2\}$, $W_{Q^V}$ and $W_{K^L}$ are learnable parameters of linear transformations. Specifically, we initialize $H_0^{refine}$ with $\hat{H}$, denoted as $H_0^{refine} = \hat{H}$. Subsequently, the cross-modality features are updated as follows,

$$
\begin{aligned}
H_j^{refine} &= H_{j-1}^{refine} + \gamma_j V^L \\
&= H_{j-1}^{refine} + \gamma_j H_{j-1}^L W_{V^L},
\end{aligned} \tag{8}
$$

where $W_{V^L}$ is a learnable parameter and $H_j^{refine}$ denotes the $j$-th ADRM layer output. After a fine-grained interaction between

visual and textual modality through ADRM, we free the feature representation from NCP and help the BART decoder to jointly learn the multimodal input.

## 3.4 Joint Learning

Following previous works, all subtasks are designed as index generation tasks and predict the token probability distribution with the BART decoder as below,

$$\tilde{H} = \lambda_2 \hat{H} + \lambda_3 H_2^{refine}, \tag{9}$$

$$h_t^d = Decoder(\tilde{H}; Y_{<t}), \tag{10}$$

$$\bar{H}^L = (E + \tilde{H}^L)/2, \tag{11}$$

$$P(y_t) = softmax([\bar{H}^L; C^d] h_t^d), \tag{12}$$

where $\lambda_2$ and $\lambda_3$ are the hyper-parameters to control the contribution from the two modules. $\tilde{H}^L$ is the textual part of $\tilde{H}$. $E$ denotes the embeddings of input tokens. $C^d$ means the embeddings of the [positive, neutral, negative, <eos>]. Finally, the loss function is defined as,

$$\mathcal{L} = -\mathbb{E}_{X \sim D} \sum_{t=1}^{O} log P(y_t | Y_{<t}, X), \tag{13}$$

where $O$ is the length of Y, and X denotes the multimodal input.

## 4 Details of the Sinkorn Algorithm

The Sinkhorn algorithm, as introduced by [6], addresses this issue by reinterpreting transport problems through a maximum-entropy perspective. It introduces an entropic regularization term to the classic OT problem, streamlining the process. This modification not only maintains the optimal solution as a meaningful distance measure but also significantly accelerates the computation. The Sinkhorn algorithm achieves this by employing matrix scaling techniques, offering a computation speed several orders of magnitude faster than traditional transport solvers.

The original Sinkhorn distance is $d_S = \langle T, S \rangle$. With the Sinkhorn algorithm, for $\lambda \in (0, \infty)$, we consider a Lagrange multiplier for the entropy constraint of Sinkhorn distances and obtain the *dual-Sinkhorn divergence* $d_S^\lambda$ as follow,

$$d_S^\lambda = \langle T^\lambda, S \rangle, \tag{14}$$

where $T^\lambda = \arg\min_{T \in \Pi} \langle T, S \rangle - \frac{1}{\lambda} h(T)$. The divergence $d_S^\lambda$ can be computed for a much cheaper cost than the original distance $d_S$. Subsequently, the optimal regularized transport $T^\lambda$ is computed as follow,

$$T^\lambda = diag(\boldsymbol{\eta}_1) P diag(\boldsymbol{\eta}_2), \tag{15}$$

where $P = e^{-\lambda S}$. $\boldsymbol{\eta}_1 \in \mathbb{R}^n$ and $\boldsymbol{\eta}_2 \in \mathbb{R}^m$ are two non-negative scaling vectors updated iteratively with Sinkhorn's fixed point iteration:

$$(\boldsymbol{\eta}_1, \boldsymbol{\eta}_2) \leftarrow (\boldsymbol{\mu}./(P\boldsymbol{\eta}_2), \boldsymbol{\nu}./(P^\top \boldsymbol{\eta}_1)). \tag{16}$$

Finally, we get optimal transport distance through $\langle T^\lambda, S \rangle$.

|  | Twitter2015 | | | Twitter2017 | | |
|---|---|---|---|---|---|---|
|  | Train | Dev | Test | Train | Dev | Test |
| Positive | 928 | 303 | 317 | 1508 | 515 | 493 |
| Neutral | 1883 | 670 | 607 | 1638 | 517 | 573 |
| Negative | 368 | 149 | 113 | 416 | 144 | 168 |
| Total Aspects | 3179 | 1122 | 1037 | 3562 | 1176 | 1234 |
| Sentences | 2101 | 727 | 674 | 1746 | 577 | 587 |

**Table 1: Statistics of Twitter2015 and Twitter2017.**

## 5 Experiments

### 5.1 Settings

**Datasets and Metrics** Following previous works, Twitter2015 and Twitter2017 [34] are taken as testbeds.The statistics of the two datasets are shown in Table 1. As for evaluation metrics, we adopt *Micro-F1* score (*F1*), *Precision* (*P*), and *Recall* (*R*) on both JMASA and MATE tasks, while on MASC task we use *Accuracy* (*Acc*) and *F1*.

**Implementation Details** Our model is developed on the foundation of BART [14]. Post an extensive hyper-parameter tuning on the development set, we fixed these parameters for consistency in our experiments. We conducted fine-tuning on downstream tasks over 35 epochs, i.e., JMASA, MATE, and MASC. The model operates with a batch size of 16 and a learning rate of 7e-5. The hidden size is maintained at 768, aligning with the BART model's specifications. Additionally, the tradeoff hyper-parameters, $\lambda_1$, $\lambda_2$ and $\lambda_3$, are set at 0.5, 1 and 0.5, respectively.

### 5.2 Comparative Baselines

In our experiments, we compare our proposed model with different methods on three types of tasks.

**Methods for Textual ABSA** 1) SPAN [10] aimed to identify opinion targets along with their corresponding sentiments. 2) D-GCN [4] effectively modeled the dependency relations among words by utilizing a dependency tree structure. 3) BART [32] addressesed seven ABSA subtasks within a unified framework.

**Methods for MATE** 1) RAN [28] focused on aligning text with corresponding object regions. 2) UMT [36] incorporated text-based entity span detection as a supplementary task. 3) OSCGA [30] emphasizes the alignment between visual objects and entities.

**Methods for MASC** 1) ESAFN [35] represented an entity-level sentiment analysis approach that leverages LSTM technology. 2) TomBERT [34] utilized BERT to derive aspect-sensitive textual representations effectively. 3) CapTrBERT [13] is designed to convert images into text and construct auxiliary sentences for enhanced fusion.

**Methods for JMASA** 1) RpBERT-collapse [24] proposed a model for MATE, utilizing collapsed labels to encapsulate aspect and sentiment pairs. 2) JML [12] stands as the pioneering joint model for MABSA, featuring an auxiliary cross-modal relation detection module. 3) VLP-MABSA [17] undertaked five task-specific pre-training tasks to effectively model aspects, opinions, and their alignments. 4) CMMT [33] implemented a gating mechanism to regulate the contributions of multimodal information during inter-modal interactions. 5) DTCA [38] proposed a dual-encoder transformer with

| Method | Twitter2015 | | | Twitter2017 | | |
|---|---|---|---|---|---|---|
| | P | R | F1 | P | R | F1 |
| *Text-based* | | | | | | |
| SPAN♣ | 53.7 | 53.9 | 53.8 | 59.6 | 61.7 | 60.6 |
| D-GCN♣ | 58.3 | 58.8 | 59.4 | 64.2 | 64.1 | 64.1 |
| BART♣ | 62.9 | 65.0 | 63.9 | 65.2 | 65.6 | 65.4 |
| *Multimodal* | | | | | | |
| RpBERT-collapse♣ | 49.3 | 46.9 | 48.0 | 57.0 | 55.4 | 56.2 |
| JML | 65.0 | 63.2 | 64.1 | 66.5 | 65.5 | 66.0 |
| VLP-MABSA♣ | 65.1 | 68.3 | 66.6 | 66.9 | 69.2 | 68.0 |
| CMMT | 64.6 | 68.7 | 66.5 | 67.6 | 69.4 | 68.5 |
| DTCA | 67.3 | 69.5 | 68.4 | 69.6 | **71.2** | 70.4 |
| AoM | 67.9 | 69.3 | 68.6 | 68.4 | 71.0 | 69.7 |
| **ADAR (Ours)** | **70.0**† | **71.5**† | **71.2**† | **71.6**† | 71.0 | **71.4**† |

**Table 2: Results comparison for JMASA. We report the average results of three runs with different random seeds. ♣ denotes the results from Ling et al. [17]. The best results are in bold and the second best ones are underlined. † denotes our model significantly outperforms baselines with $p < 0.05$ under t-test.**

| Methods | Twitter2015 | | | Twitter2017 | | |
|---|---|---|---|---|---|---|
| | P | R | F1 | P | R | F1 |
| RAN♣ | 80.5 | 81.5 | 81.0 | 90.7 | 90.7 | 90.0 |
| UMT♣ | 77.8 | 81.7 | 79.7 | 86.7 | 86.8 | 86.7 |
| OSCGA♣ | 81.7 | 82.1 | 81.9 | 90.2 | 90.7 | 90.4 |
| JML♣ | 83.6 | 81.2 | 82.4 | 92.0 | 90.7 | 91.4 |
| VLP-MABSA♣ | 83.6 | 87.9 | 85.7 | 90.8 | 92.6 | 91.7 |
| CMMT | 83.9 | **88.1** | 85.9 | 92.2 | **93.9** | 93.1 |
| AoM | 84.6 | 87.9 | 86.2 | 91.8 | 92.8 | 92.3 |
| **ADAR (Ours)** | **86.5**† | 88.0 | **87.9**† | **93.0**† | **93.9**† | **93.8**† |

**Table 3: Results comparison for MATE. We report the average results of three runs with different random seeds. ♣ denotes the results from Ling et al. [17]. The best results are in bold and the second best ones are underlined. † denotes our model significantly outperforms baselines with $p < 0.05$ under t-test.**

| Methods | Twitter2015 | | Twitter2017 | |
|---|---|---|---|---|
| | ACC | F1 | ACC | F1 |
| ESAFN | 73.4 | 67.4 | 67.8 | 64.2 |
| TomBERT | 77.2 | 71.8 | 70.5 | 68.0 |
| CapTrBERT | 78.0 | 73.2 | 72.3 | 70.2 |
| JML | 78.7 | - | 72.7 | - |
| VLP-MABSA | 78.6 | 73.8 | 73.8 | 71.8 |
| CMMT | 77.9 | - | 73.8 | - |
| AoM | 80.2 | 75.9 | 76.4 | 75.0 |
| **ADAR (Ours)** | **81.3**† | **77.1**† | **77.2**† | **76.6**† |

**Table 4: Results comparison for MASC. We report the average results of three runs with different random seeds. ♣ denotes the results from Ling et al. [17]. The best results are in bold and the second best ones are underlined. † denotes our model significantly outperforms baselines with $p < 0.05$ under t-test.**

cross-modal alignment on two auxiliary tasks to enhance performance. 6) AoM [41] designed an aspect-aware attention module and an aspect-guided GNN to detect aspect-relevant multimodal contents.

## 5.3 Main Results

The results of our ADAR and competitive baselines for the JMASA, MATE and MASC are shown in Table 2, 3 and 4 respectively, from which we have the following observations:

**Results for JMASA** (i) It is obvious that our proposed ADAR surpasses all text-based models, primarily due to richer information from different modalities. (ii) In terms of multimodal approaches, AoM outstrips previous methods on Twitter2015, largely attributable to its aspect-aware attention module. On Twitter2017, the DTCA, with its dual auxiliary tasks of text-only extraction and text-patch alignment via optimal transport, enhances cross-attention performance and proves more effective. However, we focus on aspect-driven transport and assignment instead of contrastive loss. Among all evaluated methods, ADAR demonstrates superior performance across both datasets, albeit with a marginal 0.2 absolute percentage point deficit in *Recall* compared to DTCA on Twitter2017. This notable performance gain is largely attributed to our aspect-driven alignment and refinement method, which skillfully aligns disparate modalities using aspect-driven optimal transport and augments their synergy through a dual-layer refinement process.

**Results for MATE and MASC** Table 3 and Table 4 present the outcomes for the MATE and MASC evaluations, respectively. Mirroring the pattern observed in the JMASA task, it is evident that our ADAR approach consistently delivers superior performance on both datasets, with the sole exception being the *Recall* metric for Twitter2015. These findings further underscore the broad efficacy of our methodology.

## 5.4 Ablation Study

To validate the efficacy of each component in ADAR, we conducted a set of ablation experiments on JMASA task, and the results are reported in Table 5.

**Effect of ADAM** On Twitter2015, disabling ADAM solely leads to a significant reduction in performance. With a 7.44% drop on *F1*, it underscores the critical role of the aspect-driven transport method in effectively aligning different modalities for a further settlement of NCP in ADRM.

**Effect of ADRM** Removing ADRM solely results in a substantial decline of 5.06% on *F1* in Twitter2015 performance. This highlights the importance of the refinement process following alignment, as it is essential for facilitating interaction between modalities and enhancing feature information richness in an aspect-driven way.

**Effect of AFB** As an auxiliary module for ADAM, ablating AFB highlights its proficiency as a simple yet effective approach to noise reduction, thereby making a beneficial contribution to the overall system's performance.

| Method | Twitter2015 | | | Twitter2017 | | |
|---|---|---|---|---|---|---|
| | P | R | F1 | P | R | F1 |
| **ADAR** | **70.0** | **71.5** | **71.2** | **71.6** | **71.0** | **71.4** |
| w/o ADAM & ADRM | 62.5 | 62.1 | 62.0 | 63.5 | 64.1 | 63.7 |
| w/o ADRM & AFB | 65.2 | 66.7 | 66.3 | 66.7 | 67.0 | 66.8 |
| w/o ADRM | 67.0 | 67.8 | 67.6 | 67.9 | 68.2 | 68.0 |
| w/o ADAM | 64.6 | 66.0 | 65.9 | 65.9 | 66.5 | 66.0 |
| w/o AFB | 68.8 | 70.2 | 69.6 | 69.1 | 69.9 | 69.5 |

**Table 5: Results of ablation experiments.**

## 5.5 Hyper-parameter Analysis

**Layer Number of ADRM**  We carried out a series of experiments to ascertain the optimal number of layers for the Aspect-driven Refinement (ADRM) module. The performance outcomes associated with varying layer counts are detailed in Table 6. With a single ADRM layer, we observed comparatively weaker performance, attributed to insufficient fitting of cross-modality features. Conversely, employing three ADRM layers resulted in overfitting, paradoxically leading to lower performance than with a single layer. Consequently, in light of our comparative analysis, we have opted for a two-layer configuration for the ADRM module.

**Hyper-parameter $\lambda_1$ and $q$**  To effectively leverage the aligned information following optimal transport, we conducted experiments with different settings of the hyper-parameter $\lambda_1$ and $q$. As depicted in Figure 4, a peak at $\lambda_1 = 0.5$ is observed in both datasets. The performance is weak due to an underutilization of aligned features at $\lambda_1 = 0.25$. As $\lambda_1$ increases beyond the optimal value, there is also a performance decline, due to the introduction of noise through excessive reliance on optimal transport. To mitigate this, we implement an adaptive filter bin to eliminate irrelevant information with a threshold $q$. The model achieves the best performance on both datasets when $q$ is set to 0.2. We observe that setting $q$ higher than 0.2 leads to a decline in performance, mainly due to mistakenly filtering of relevant information. It is also worth noting that the performance on the Twitter2017 dataset was not as strong as on Twitter2015 when $q$ is set to 0.4 or 0.5. This can be attributed to the greater number of aspects per image-text pair in Twitter2017. When combined with a strict filtration policy, it results in reduced effectiveness due to the increased interference among aspects. The above analysis demonstrates the effectiveness of our proposed coarse-to-fine aspect-driven alignment module.

| # Layer number | Twitter2015 | | | Twitter2017 | | |
|---|---|---|---|---|---|---|
| | P | R | F1 | P | R | F1 |
| 1 | 66.1 | 67.5 | 67.4 | 68.2 | 69.0 | 68.5 |
| 2 | **70.0** | **71.5** | **71.2** | **71.6** | **71.0** | **71.4** |
| 3 | 65.2 | 66.7 | 66.3 | 66.7 | 67.0 | 66.8 |

**Table 6: Performance comparison of different layer numbers of ADRM for JMASA.**

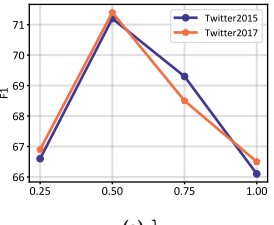
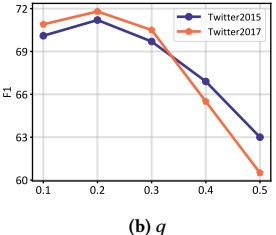

(a) $\lambda_i$      (b) $q$

**Figure 4: *F1* comparisons of different hyper-parameters $\lambda_1$ and $q$ on JMASA.**

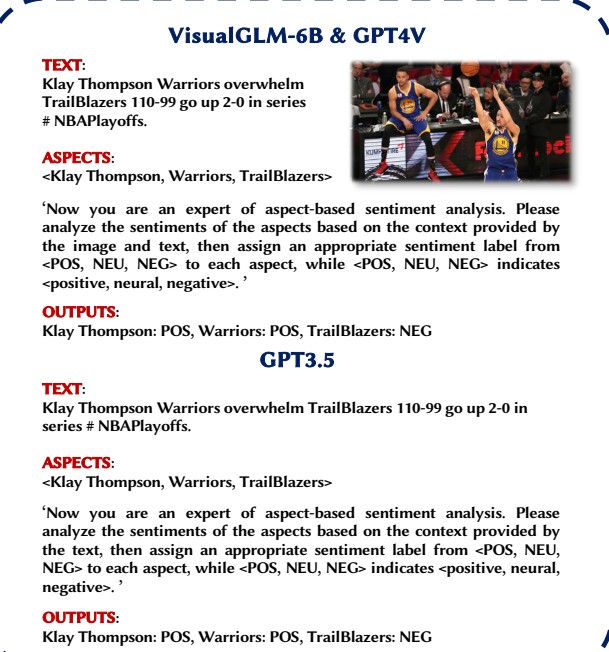

**Figure 5: An illustration of prompts for VisualGLM-6B, GPT3.5 and GPT4V.**

## 5.6 Comparison with LLMs on MASC

In recent advancements, Large Language Models (LLMs) have demonstrated remarkable capabilities in various Natural Language Processing (NLP) tasks, leveraging their advanced language understanding and generation skills [15, 26]. To ascertain the competitive edge of our model on MABSA, comparative analyses were conducted against prevalent LLMs, including VisualGLM-6B, GPT3.5, and GPT4V. Considering the inherent design limitations of LLMs in aspect identification and output structures, we confined the comparison to the MASC task to maintain evaluative fairness. To conduct the experiments, we design two prompt templates. As shown in Figure 5, we setup the role for LLMs and define the task. Subsequently, we provide the image, text and aspects to acquire the outputs. Specifically, when experimenting with GPT3.5, we provide

| Models | Twitter2015 | | Twitter2017 | |
|---|---|---|---|---|
| | ACC | F1 | ACC | F1 |
| **ADAR (ours)** | **81.3** | **77.1** | **77.2** | **76.6** |
| VisualGLM-6B | 66.1 | 68.2 | 69.0 | 68.5 |
| GPT3.5 | 65.2 | 66.7 | 67.0 | 66.8 |
| GPT4V | 75.3 | 74.2 | 76.0 | 75.5 |

**Table 7: Comparison with LLMs on MASC.**

only the text and aspects, owing to its absence of multimodal capabilities. The comparative results, as depicted in Table 7, demonstrate the superior performance of our ADAR, which achieves notable performance despite utilizing fewer parameters than LLMs.

It is imperative to acknowledge the constraint of GPT3.5 that it solely processes textual inputs. Despite its commendable generative prowess, GPT3.5's performance in multimodal tasks was comparatively subdued. This outcome reveals the significant potential and necessity of integrating and leveraging cross-modal information to enhance the accuracy and applicability in multimodal NLP tasks. Thus, GPT4V shows a better performance on MASC with its powerful multimodal capability. However, our ADAR with fewer parameters and training time, while not reaching the same level of complexity or depth as GPT4V, demonstrates a notable efficiency on MASC.

### 5.7 Error Analysis

To better learn and improve future work on MABSA, we conduct experiments on accuracy of sentences with different characteristics for JMASA on Twitter2015 and Twitter2017 test set. As shown in Table 8, the two test set are different in aspects and sentiments distribution. Twitter2015 focus on one aspect and single sentiment circumstance, while Twitter2017 contains more sentences with multiple aspects and sentiments. From Figure 6, we observe that the trend appears to be similar for all three methods in both datasets. The accuracy is higher when only one aspect is mentioned and tends to decrease as the complexity of the sentence increases with multiple aspects and multiple sentiments. For Twitter2015, our ADAR generally outperforms DTCA and AoM across all three characteristics, indicating it handles complexity better due to the aspect-driven method. For Twitter2017, the same pattern holds, although the differences in accuracy between ADAR and the other two methods are less pronounced. ADAR shows a consistent performance advantage over the other two methods, which indicates it has a better handling of sentences with multiple aspects and sentiments.

| | Twitter2015 | Twitter2017 |
|---|---|---|
| # sentences | 674 | 587 |
| # one aspect | 416 (61.72%) | 188 (32.03%) |
| # multiple aspects | 258 (38.28%) | 399 (67.97%) |
| # multiple sentiments | 104 (15.43%) | 263 (44.80%) |

**Table 8: Detailed statistics of two test set. # x denotes the count of sentences with the specified characteristic x.**

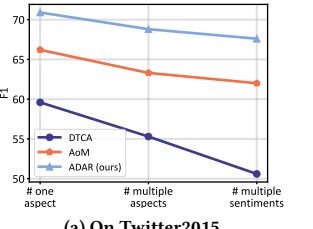 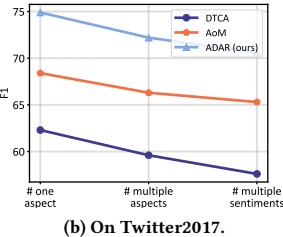

(a) On Twitter2015.                    (b) On Twitter2017.

**Figure 6: Accuracy of sentences with different characteristics for JMASA on Twitter2015 and Twitter 2017.**

### 5.8 Computation Efficiency

Additionally, we experimented to analyze our method's computation efficiency compared to SOTA models. As shown in Table 9, our method outperforms AoM and approaches VLP-MABSA in inference time per step due to the Sinkhorn algorithm we use. As described in Section 4, this algorithm reinterprets transport problems through a maximum-entropy perspective and introduces an entropic regularization term to the classic OT problem, streamlining the process. Thus, while our method's complexity only lies in the theoretical aspects of optimal transport, it is easy and efficient to apply in practice.

| | Latency/Inference Time per Step |
|---|---|
| VLP-MABSA | 0.484s |
| AoM | 0.563s |
| Ours | 0.487s |

**Table 9: Comparison with SOTA baselines on computation (inference) latency.**

### 6 Conclusion

In this paper, we proposed the Aspect-driven Alignment and Refinement model crafted for the task of MABSA, which seamlessly unifies the Coarse-to-fine Aspect-driven Alignment module and the Aspect-driven Refinement module. Notably, our model addresses NCP from the perspective of aspects. Empirical analysis on two extensively utilized datasets substantiates the efficacy of our approach.

### Limitations and Future Work

Despite the notable superiority of our proposed method over existing SOTA approaches, it is imperative to acknowledge and address several challenges in future research endeavors. Firstly, predicting emotions of tweet posts, which are related to contemporary issues, is hindered by a lack of requisite external knowledge. Secondly, our ADAM method exhibits limitations in dealing with missing visual elements. Acting as a double-edged sword, the adaptive filter bin effectively removes irrelevant multimodal entities but also overlooks the missing information. These limitations present a critical area for further investigation in our subsequent research efforts.

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
