# OpenReview forum: "Aspects are Anchors: Towards Multimodal Aspect-based Sentiment Analysis via Aspect-driven Alignment and Refinement"
_acmmm.org/ACMMM/2024/Conference — MM2024 Poster_

### Official Review · Reviewer_3MwE · 2024-05-14

**Rating:** 3
**Confidence:** 3

**Summary:**

This paper focuses on the noisy correspondence problem (NSP) in the multimodal aspect-based sentiment analysis task and proposes an Aspect-Driven Alignment and Refinement framework. It first devises a coarse-to-fine aspect-driven alignment module to learn the alignment between visual and textual features. Then, an aspect-driven refinement module is used to refine the feature representation. Experiments on two benchmarks across 3 tasks show that the proposed method outperforms the other counterparts.

**Strengths:**

-	This paper focuses on an interesting problem and proposes a new method to solve it.
-	The paper is overall well-written and the storyline is clear.
-	Extensive experiments show the effectiveness of the proposed method.

**Limitations:**

-	The proposed framework is only evaluated on the BART-based models. The paper will be more convincing if the framework can work well on the other LLMs or MLLMs.
-	The compared methods are weak, as the newest compared method is AoM published in ACL2023. I suggest to compare with more recent counterparts, e.g., “WisdoM: Improving Multimodal Sentiment Analysis by Fusing Contextual World Knowledge” (arXiv 2024).
-	Some claims are not well confirmed by the experiments. For example, there is a lack of experiments and analyses to investigate whether the proposed methods can effectively alleviate the NSP problem.
-	The proposed methods are relatively complex. More experiments on the efficiency of proposed methods are required.

**Suitability:**

2

---

### Official Review · Reviewer_u3Tf · 2024-05-23

**Rating:** 5
**Confidence:** 2

**Summary:**

This paper mainly addresses the problem of poor alignment between visual content and text description in Multimodal Aspect-based Sentiment Analysis tasks. This article proposes a two-stage framework that introduces the Coarse to fine Aspect driven alignment module to learn coarse-grained alignment between text and vision, and the Aspect driven Refinement module to refine multimodal feature representations. This article conducted a series of experimental verifications on the proposed framework, and the experimental verification results showed the effectiveness of the framework.

**Strengths:**

This paper is innovative to some extent, and it is the first work to introduce Optimal Transport into MABSA to solve the NCP problem.
The experimental section of this article is rigorous, sufficient, and complete. This article conducts experiments on two datasets, Twitter2015 and Twitter2017, using multiple evaluation metrics such as ABSA, MATE, MASC, and JMASA, and compares them with current advanced models to verify that the proposed method has good performance. At the same time, ablation experiments were conducted to verify the effectiveness of each module. This article also compared it with the currently popular LLM and found that its performance can even surpass LLM with fewer parameters. This article also analyzes the errors in framework prediction, providing ideas for future work.

**Limitations:**

The adaptive filter bin proposed in this article may filter out some useful information, causing the model to move from one side to another, which may decline the performance of the model.

**Suitability:**

3

---

### Official Review · Reviewer_nxEa · 2024-05-24

**Rating:** 5
**Confidence:** 3

**Summary:**

This paper introduces the ADAR model to perform MABSA using a two-stage coarse-to-fine alignment framework. The idea is interesting, but some issues need to be addressed.

**Strengths:**

Give a new framework for the MABSA, the idea is good and interesting.
The paper's writing is good with the proper organization.

**Limitations:**

Initial Word Embedding Impact: Does the initial word embedding affect the experimental results? Specifically, did the compared models (e.g., those mentioned in references [14] and [32]) also use the same initial word embeddings or visual embeddings?
 There is no evaluation metric provided to demonstrate how well the alignment between image and text is achieved. This is important since the paper claims to have a superior alignment framework.
The improvement shown in the experiments seems limited compared to the current state-of-the-art methods.

**Suitability:**

3

---

### Official Review · Reviewer_BrNB · 2024-06-04

**Rating:** 4
**Confidence:** 3

**Summary:**

In this paper authors propose a method named Aspect-driven Alignment and Refinement (ADAR) for Multimodal Aspect-based Sentiment Analysis (MABSA). In particular, this framework is able to extract scene subjects and perform sentiment analysis for each subject based on a picture of the scene and on a descriptive caption.
The method differs from previous works in some aspects, such as the adoption of a dedicated alignment module based on Optimal Transport to tune the alignment between textual and visual features and an additional refinement module to further refine cross-modal features.
The proposal is evaluated on two datasets, demonstrating improvements in terms of Precision, Recall and F1 compared to existing works. An extensive configuration and ablation study is proposed to evaluate the impact of each choice made in the experiments.

**Strengths:**

- The paper is well written
- Experimental evaluation is vast and deeply detailed
- I appreciate the addition of unconventional but interesting sections that unveils useful insights, such as the hyper-parameter impact analysis and the comparison with general-purpose LLMs.

**Limitations:**

- In Table 2, I assume "P", "R" stand for Precision and Recall. I understand they may have been contracted for spacing reasons but a legend in the caption is necessary for clarity.

- The performance gains stand in the range of 1-2% compared to top-performing baselines, which is not an impressive result compared to the complexity of the method.

I find the paper to be in-theme with the conference and the work is worth publishing. However, I am not convinced by the quantitative improvements, especially when we compare it with the complexity introduced in the framework and when we consider that the most modern dataset dates back to 2017.

**Suitability:**

3

---

### Meta-Review · Area_Chair_9h29 · 2024-07-02

**Recommendation:** Accept (Poster)
**Confidence:** 5

**Metareview:**

The paper received mixed scores. The average score is on the positive side, however most of positive scores did not consider the rebuttal. In general, there still are concerns.